# Effect of Yeast Culture (*Saccharomyces cerevisiae*) on Broilers: A Preliminary Study on the Effective Components of Yeast Culture

**DOI:** 10.3390/ani10010068

**Published:** 2019-12-30

**Authors:** Zhe Sun, Tao Wang, Natnael Demelash, Sen Zheng, Wei Zhao, Xue Chen, Yuguo Zhen, Guixin Qin

**Affiliations:** 1College of Life Science, Jilin Agricultural University, Changchun 130118, China; sunzhe198615@163.com; 2JLAU-Borui Dairy Science and Technology R & D Centre, Jilin Agricultural University, Changchun 130118, China; cagewang@163.com (T.W.); 13756988604@163.com (W.Z.); chenxue@borui.com (X.C.); 3College of Animal Science and Technology, Jilin Agricultural University, Changchun 130118, China; ednatnael@gmail.com (N.D.); kefeng4469@outlook.com (S.Z.)

**Keywords:** yeast culture, broiler, metabolomics, gas chromatography-mass spectrometry, multivariate analysis

## Abstract

**Simple Summary:**

The value of yeast culture (YC) as alternative feed additives in poultry farming has been proven. YC is a nutrient-rich and complex micro-ecological fermentation product containing various metabolites. However, the major or specific effective components of YC and their importance in poultry farming are unknown. Herein, we screened the “effective ingredients” of YCs obtained from different fermentation times based on metabolomics and animal feeding experiments. Glycine, fructose, inositol, galactose, and sucrose were identified as potential effective metabolites in YCs. These findings provide an important basis for objective, accurate, and quick evaluation of the quality of YC products, as well as a scientific understanding of their functions.

**Abstract:**

This study was aimed at determining the effective ingredients of yeast culture (YC) for animal breeding. First, the contents of YCs obtained from various fermentation times were detected using gas-chromatography. A total of 85 compounds were identified. Next, 336 Arbor Acres (AA) broilers were randomly divided into seven experimental groups and fed a basal diet, diets supplemented with YCs obtained at various fermentation times, or SZ1 (a commercial YC product). A significant increase in body weight gain (BWG) and a significant decrease in feed conversion ratio (FCR) of AA broiler chicks were observed with YC supplementation. Additionally, most of blood and immunological indices were improved with YC supplementation. According to the production performance and the results of multivariate analysis, glycine, fructose, inositol, galactose, and sucrose were found as the potential effective compounds of YC and were involved in metabolic pathways including glycine, serine, and threonine metabolism. Supplementation with diets based on combinations of effective compounds improved weight gain, feed efficiency, serum immunoglobulin A, and immunoglobulin G, but decreased blood urea concentration. These findings suggest YCs as effective and harmless feed additives with improved nutritional properties for broiler chicks.

## 1. Introduction

The growing requirements for safe and good quality poultry products have led to an increased interest in growth-promoting feed additives for animal nutrition [1,2,3]. Considerable research has been conducted to evaluate the potential growth performance and health benefits of yeast-based products for animals [4]. Previous studies indicated that supplementation with yeast culture (YC) enhances animal nutrition and health, and that dietary supplementation of YC (*Saccharomyces cerevisiae*) could increase the body weight and feed efficiency, and promote the immune system in animals, while decreasing the percentage of abdominal fat in broilers [5,6]. It is important to analyze the metabolites in yeast culture because some metabolites in yeast culture are beneficial for stimulating bacterial growth in the digestive tract and optimizing animal feed intake. However, the exact composition of YC and the efficiency of its components are not well elucidated, and need an in-depth investigation.

YC is a unique micro-ecological product that contains a combination of yeast biomass and fermentation metabolites produced during the fermentation processes. While it is mainly composed of yeast extracellular metabolites such as peptides, alcohols, esters, and organic acids [7], the composition of the medium changes after fermentation and a number of residual viable yeast cells persist. Studies indicate that YC nutritional and health care function is attributable to its fermentation metabolites, which comprise growth factors such as pro-vitamins and/or micronutrients that stimulate animal growth [8,9]. However, the composition of the various metabolites produced depends on the composition of the fermentation media used and the fermentation conditions. The safety of these metabolites is unknown and their main or specific activity and potential use have not been identified and functionally assessed. Furthermore, there is no standard and accurate analytical method for estimating biologically important chemical components of YC products, and limited research has been conducted on this aspect in the feed industry. Owing to the increasing interest in using YC products in animal feeds, the development of analytical approaches to estimate YC and its effective components is much needed to ensure the quality of animal products.

Metabolite profiling in YCs can be performed by a wide range of analytical techniques. Because of its sensitivity, low detection limit, selectivity, and small sample size requirement, most yeast metabolites are analyzed by gas chromatography (GC). Gas chromatography-mass spectrometry (GC-MS) can be used to analyze various derivatives of non-volatile and volatile metabolites. However, the component analysis of YC is based on the GC-MS metabolomics method, which can produce data containing a large number of molecules. It is difficult to find the similarities and differences between samples or groups by conventional statistical analysis methods, and it is also difficult to find out which molecules in the sample contribute to the discrepancies. Therefore, in general, multivariate analysis, clustering analysis, neuron network analysis, and other methods are used, with multivariate analysis being the most widely used [10]. Multivariate analysis includes principal component analysis (PCA), partial least squares discriminant analysis (PLS-DA), and orthogonal partial least squares discrimination analysis (OPLS-DA). PCA classifies a large amount of original spectrogram information or pre-processed data information into samples, characterizes them by certain visualization methods, and screens out biomarkers closely related to the original information. PLS-DA and OPLS-DA are methods that are applied for establishing mathematical models of multi-dimensional complex sample groups to maximize separation, and to establish a multi-parameter modelling model for predicting unknown samples. At present, the commonly used data analysis method is the PCA method, which can visualize the grouping of the original samples and help establish the visualization model by the PLS-DA or OPLS-DA method [11,12,13]. The application of these multivariate analysis methods could be useful in identifying and predicting the effective compounds of YCs.

Hence, the main objectives of this study were to (1) determine the effective ingredients of YC using metabolomics and (2) test the effectiveness of these ingredients as feed additives for broilers.

## 2. Materials and Methods

The animal care protocol was approved by the Jilin Agricultural University Animal Care and Use Committee and Commercial Chicken Farm (JLAU-ACUC2017-006, Changchun, China).

### 2.1. Preparation of YCs

A strain of *Saccharomyces cerevisiae* (No. 2012), screened in the laboratory from the JLAU-Borui Dairy Science and Technology R&D Centre of the Jilin Agricultural University (Changchun, China), was used in this study. *S. cerevisiae* was aerobically cultured in medium containing 10% molasses, 10% brown sugar, 0.5% urea, 0.5% yeast powder, 0.05% magnesium sulfate, and 0.05% glycine, and incubated for 24 h. Then, papain was added to induce cell wall breakage, followed by anaerobic fermentation for 12, 24, 36, 48, and 60 h. The culture broths were collected at indicated fermentation times and freeze-dried to obtain the different YCs (12YC, 24YC, 36YC, 48YC, and 60YC). The freeze-dried YCs were used for subsequent experiments.

### 2.2. Metabolomics Analysis of YCs

#### 2.2.1. Metabolites Derivatization

The yeast samples (100 μL) of each group were mixed with 300 μL of methanol/chloroform (3:1). After 10 min standing at −20 °C, the mixed liquor was centrifuged at 12,000× *g* at 4 °C for 10 min, and then 300 μL of the supernatant was transferred to the test tubes. The yeast samples were dried using the freeze drier for 48 h. The drying samples of yeast (0.05 g drying samples) were stored in 1.5 mL tubes. One hundred microliters of 20 mg/mL methoxyamine hydrochloride was added to the yeast samples, followed by heating in a water bath at 37 °C for 90 min. Subsequently, 200 µL bis(trimethylsilyl)-trifluoroacet-amide (BSTFA) with 1% trimethylchlorosilane (TMCS) (Sigma-Aldrich, Castle Hill, Australia) heated at 70 °C for 60 min was added to the dried extract of yeast samples to complete the derivatization process. After the derivatization, the samples of each group were centrifuged at 12,000× *g* at 4 °C for 10 min. Then, 50 mL of the supernatant was collected and, after adding 0.5 mL of n-hexane, samples were injected directly into the gas chromatography-mass spectrometer (GC-MS) for analysis. Under the same analysis conditions, samples of each group were repeated six times.

#### 2.2.2. Metabolites Identification by GC-MS

The analysis of yeast samples was performed using GC-MS (7890A/5975C; Agilent Inc, Santa Clara, CA, USA). The electron impact (EI) ionization mode used was at 70 eV. A 30 m HP-5MS capillary column with an internal diameter of 250 μm and a film thickness of 0.25 μm was used. All injections were done in the split less mode with 1 μL injected volume, and an oven ramp beginning at 80 °C (hold for 3 min), then increasing at a program rate to 150 °C with a hold time of 10 min. Helium (carrier gas) was used at a rate of 1.0 mL/min. The transfer line was maintained at 280 °C with an acquisition rate of 10 Hz. The shunt ratio was 10:1, the solvent delay time was 3 min, and mass scanning range was 20–800 amu.

#### 2.2.3. Screening of Differentially Expressed Metabolites

All metabolomics data were analyzed using SIMCA-P 14.0 (Umetrics AB, Umeå, Sweden). Multivariate analysis included principal component analysis (PCA), partial least squares discriminant analysis (PLS-DA), and orthogonal partial least squares discrimination (OPLS-DA). The first principal component of the PLS (variable importance projection, VIP value) was obtained to identify the differentially expressed metabolites. Molecules with VIP values exceeding 1.0 and *p*-values less than 0.05 were selected as differentially expressed metabolites.

#### 2.2.4. Pathway Characterization

The compound names and relative concentrations of the shared metabolites were imported into Metaboanalyst 3.0 (http://www.metaboanalyst.ca) to perform functional enrichment and impact pathway analysis. Impact values greater than 0.10 and *p*-values less than 0.05 were defined as significant impact pathways. Extremely different pathways were defined with *p*-values less than 0.01. In addition, databases such as the Kyoto Encyclopedia of Genes and Genomes (KEGG, http://www.genome.jp/kegg/) and the Small Molecular Pathway Database (SMPDB, http://smpdb.ca) were used to search for metabolites and for the integrated pathway analysis.

### 2.3. Preparation of Diets

The ingredients and nutrient composition of diets fed during the starter (1–21 days of age) and finisher (22–42 days of age) periods are reported in Table 1 as described in our previous study [14]. For each treatment group, different YCs were added to the basal diet at a dosage of 0.192% (dry weight of the fermentation broth/feed (g/g)) and subsequently mixed and stirred with a mixer. The YC was in the form of powder and the inclusion rate was chosen based on the results of our preliminary experiments. All diets were antibiotic free and were formulated to meet the nutrient requirements of the basal diet for Arbor Acres (AA) broilers. To test the effectiveness of YC effective substances, these compounds were commercially acquired and used for preparing different combinations (A, B, and C groups are the combination groups). The proportions of these compounds were predicted using a linear regression model. The combination diets were prepared by mixing the pure powder of each compound with the basal diet.

### 2.4. Animal Production Trials

#### 2.4.1. Trial 1

Three hundred and thirty-six one-day-old, gender-mixed AA broiler chicks obtained from a local commercial hatchery were used in this experiment. Chicks were randomly distributed into 42 cages (seven groups × six replications; each replication included eight chicks). Each group was allocated to one of the seven dietary treatments. One of the treatments was a corn–soybean basal diet with no addition (control), while the other treatments were done with the basal diet supplemented with YC from different fermentation times: 12 h (YC12), 24 h (YC24), 36 h (YC36), 48 h (YC48), and 60 h (YC60), as well as SZ1 (a commercial YC product). The diets were given from 0 to 6 weeks.

#### 2.4.2. Trial 2

On the basis of the performance results of trial 1 and preliminary screening results of the active ingredients of the YCs, the presence of active ingredients was confirmed. A combination of these active components was determined according to the established mathematical model of the active ingredients and the production performance. Trial 2 was conducted to test the efficacy of the active components of YC as an additive feed in broiler nutrition and to determine its effectiveness as a supplement. Two hundred and twenty-eight one-day-old, gender-mixed AA broiler chicks obtained from a local commercial hatchery were used in this experiment. Chicks were randomly distributed into 36 cages (six groups × six replications, each replication included eight chicks). Each group was allocated to one of six dietary treatments. The control treatment was a corn–soybean basal diet with no addition (as in trial 1). The A, B, and C groups were fed the basal diet supplemented with active ingredients of YC in different combinations. Animals in the YC24 and SZ1 groups were fed YC24 or SZ1, respectively, as supplements. Broilers were housed in the same condition as in trial 1. The experiments lasted for six weeks.

#### 2.4.3. Growth Performance

Eights birds per pen were weighed together each week to determine the live body weight and weight gain. Feed conversion ratio (FCR) was calculated as the ratio of feed intake to body weight gained. The body weight gain (BWG), feed intake (FI), and feed conversion ratio (FCR) were also calculated for each growing phase as follows: from weeks 0 to 3, weeks 4 to 6, and weeks 0 to 6 for the entire six weeks of the experimental study. Mortality was checked over the production trial period and no additional adjustment was performed because no death of broilers was recorded.

#### 2.4.4. Blood Sample Collection and Analyses

Before blood sample collections, feed was removed from all birds for a period of 3 h in an attempt to allow stabilization of blood constituents. On the morning of day 21 and day 42, a 3.0 mL venous blood sample was collected from the ulnaris vein of one bird from each treatment into tubes containing 1.8 mg/mL of spray-dried K2 ethylenediaminetetraacetic acid (EDTA) (Greiner Bio-One GmbH, Kremsmunster, Austria). Blood samples were centrifuged at 3000× *g* for 15 min to assure separation of the blood cells and then stored at 20 °C for further analysis of total protein (TP), alanine aminotransferase (ALT), alkaline phosphatase (ALP), aspartate aminotransferase (AST), and blood urea nitrogen (BUN) using the Autoanalyzer Medical System (Autolab, BT 3,500 Autoanalyzer Medical System, Rome, Italy), and were measured using reagent kits (Wako Pure Chemical Industries, Osaka, Japan). Serum immunoglobulin G (IgG) and immunoglobulin A (IgA) were measured using the double-antibody sandwich enzyme-linked immunosorbent assay (ELISA) with commercial kits (Bethyl Laboratories, Montgomery, TX, USA).

### 2.5. Statistical Analysis

Data were analyzed using SPSS 23.0. The data were compared using one-way analysis of variance (ANOVA) and significant differences between groups were analyzed by Duncan’s multiple comparisons; significance was declared at *p* < 0.05. The means, pooled standard error of the mean (SEM), and *p*-values are summarized in tables. The correlation analysis between the active ingredient content of the selected YC and BWG was performed, and the stepwise regression method was used to establish the prediction equation that was used for predicting the proportion of key metabolites in combination diets.

## 3. Results

### 3.1. Analysis of YC Components

The results of metabolites identification of the YCs are shown in Table 2. In total, 85 compounds were identified from all of the YCs obtained at various fermentation time points. The composition of the YCs differed from one to another, and metabolites specific to some YCs were identified (Table 2). Common metabolites were also identified, but their concentrations (peak areas) differed from one YC to another (Table 2).

### 3.2. Trial 1

#### 3.2.1. Growth Performance

The effect of the basal diet supplemented with YCs from different fermentation periods on the performance of broilers (0–6 weeks) in trial 1 is presented in Table 3. There was no significant difference in the initial body weight (BW) of broilers attributed to each feeding group (*p* > 0.05), but significant differences in BW were found at day 21 and day 42 of animal production trials (Table 3). At day 21, the feeding with diets supplemented with YC12, YC24, and YC36 led to increased BW of broilers compared with the control group, but the supplementation with YC48, YC60, and SZ1 led to insignificant changes (Table 3). At day 42, supplementation with YC12, YC24, YC36, YC48, and SZ1 increased BW of broilers, whereas the supplementation with YC60 did not significantly impact the BW compared with the control group (Table 3).

The body weight gain (BWG) was calculated over a different interval of time. During weeks 0–3, the supplementation with YC24 significantly increased BWG (*p* < 0.05) relative to the control group, whereas no significant difference was found between the control group and the YC12, YC36, YC48, YC60, and SZ1 groups. No significant difference was observed among the YC12, YC24, YC36, and SZ1 groups. No significant change in FI was found between the groups (Table 3). However, FI was significantly increased in the YC36 group compared with the control group (Table 3). Moreover, feeding with the YC supplement led to decreased FCR, except with a significant difference for YC24 (*p* < 0.05), while no obvious difference was found between the control group and the other diets.

During weeks 4–6, BWG was significantly increased for broilers fed with diets supplemented with YC12, YC24, YC36, and SZ1 (Table 3). The greatest BWG (1691 g) value was obtained with YC24 (Table 3). No significant difference in BWG was found between the control group and the YC48 and YC60 groups (Table 3). The FI was increased with dietary supplementation of YC24 and YC48 (*p* < 0.05), but no significant difference was found with the YC12, YC36, YC60, and SZ1 dietary supplementations compared with the control group (Table 3). The FCR values of broilers were decreased after supplementation with YC12, YC24, YC36, and SZ1, but increased with YC48 and YC60. The lowest FCR was obtained with the YC24, which showed a significant difference from the other groups. A significant difference in FCR was also found between the control group and the YC36, YC48, and YC60 groups, whereas the FCR values in the YC12, SZ1, and control groups were not significantly different.

During weeks 0–6, BWG of broilers was increased by the supplementation with YC12, YC24, YC36, YC48, YC60, and SZ1 (Table 3). Statistical significance was recorded for YC12, YC24, YC36, YC48, and SZ1, and the highest BWG was obtained with YC24 supplementation (Table 3). For FI, an increase was also recorded and a significant increase for YC24, YC36, YC48, YC60, and SZ1 was observed. No significant difference was found between YC12 and the control group (Table 3). The FCR was decreased by dietary supplementation with YC12, YC24, YC36, and SZ1, with significant differences recorded for YC24, YC36, and SZ1 (*p* < 0.05), but there was no significant difference between the control and YC12 (Table 3). A significant increase was found for YC48 and YC60 compared with the control (Table 3).

#### 3.2.2. Biochemical Profiles

There was no significant difference in the content of total protein (TP) between the control, YC24, and YC60 groups. The YC treatment in the YC12, YC36, and YC48 groups, as well as SZ1, increased the TP content compared with the control group (*p* < 0.05, Table 4).

The enzyme activity of alkaline phosphatase (ALP) in the serum was significantly increased in the YC12–YC60 groups compared with the control group (*p* < 0.05, Table 4), while no significant difference was found between the YC and the control group. A significant increase in aspartate aminotransferase (AST) activity was observed in the YC12, YC36, and SZ1 groups compared with the control group (*p* < 0.05, Table 4), whereas no significant difference was found between the control, YC24, YC48, and YC60 groups. The activity of alanine aminotransferase (ALT) was increased upon feeding with YCs and a significant difference was found between the control group and the SZ1, YC24, YC36, YC48, and YC60 groups (*p* < 0.05, Table 4). No significant difference was found between the control group and the YC12 group. The blood urea nitrogen (BUN) level was significantly decreased in broilers fed with YC when compared with those in the control group (*p* < 0.05, Table 4). No significant difference was found between the YC12 and control groups, but YC24 supplementation caused the greatest decrease in BUN level (*p* < 0.05).

#### 3.2.3. Serum Immunoglobulins

The contents of IgA and IgG in serum are indicated in Figure 1. On day 21, the level of IgA was significantly increased following SZ1- and YC-supplemented diets compared with the control group. On day 42, compared with the control group, a significant difference in IgA level was found upon feeding with SZ1-, YC24-, YC36-, and YC60-supplemented diets. On days 21 and 42, serum IgG levels were increased (*p* < 0.05) in broilers fed the SZ1-, YC24-, YC36-, and YC48-supplemented diets as compared with broilers fed the basal diet. YC36 supplement showed the highest level of serum IgG. No significant difference was recorded between the control group and the YC60-supplemented diet. On day 42, except for YC12, a significant difference (*p* < 0.05) was found between the control group and the other experimental groups.

### 3.3. Identification of Effective Metabolites of YC

The PCA score plot of the metabolic profiles showed significantly separated clusters among the YCs (Figure 2). All score plots for the YC12, YC24, YC36, YC48, YC60, and SZ1 samples were in the Hotelling T2 ellipse with 95% confidence. The distance between the commercial product SZ1 and the other treatment groups was the greatest. In addition, YC24 and YC60 samples were far apart from each other.

On the basis of the production performance and the biochemical and immunological indices, YCs were considered to contain effective metabolites when they exhibited good performance. As YC60 showed the worst production performance and YC24 exhibited the best production performance, these two YCs were used for screening differential effective metabolites. As shown in Figure 3a, the YC24 and YC60 samples were clearly separated into two parts and all score plots for the samples in the two groups were in the 95% Hotelling T2 ellipse in the OPLS-DA score map (Figure 3a). A clear separation in the metabolomic profiles was observed between YC24 and YC60. The S-plot showing the key metabolites is depicted in Figure 3b. On the left and right side of the S-plot, the variables with strong model contribution and high statistical reliability were screened for uncovering potential biomarkers to characterize the metabolic discriminations between YC24 and YC60. In total, five differentially expressed metabolites were identified between YC24 and YC60 (very important parameter (VIP) > 1, *p* < 0.05), as indicated in Table 5.

In addition, using KEGG pathway analysis (Figure 4), we found that the differentially expressed metabolites screened between YC24 and YC60 were enriched in four significant pathways: glycine, serine, and threonine metabolism; inositol phosphate metabolism; galactose metabolism; and starch and sucrose metabolism (*p* < 0.05, impact value > 0.10).

### 3.4. Trial 2

#### 3.4.1. Growth Performance

Taking into account the results obtained in trial 1, which indicated that YC24 contained the most effective components and good performance on growth, immunity, and biochemical parameters after its supplementation, and considering that shorter fermentation time is cost-effective, a regression model was generated based on YC24 supplementation. According to the model, the active ingredients obtained from the model were commercially purchased and combined, and the combination scheme was as shown in Table 6.

The effect of supplementation of different combinations of active components in the corn–soybean basal diet on the performance of broilers in trial 2 is shown in Table 7. The initial BW values of broilers were not significantly different (Table 7). However, at day 21, the BW was significantly increased in combination diets and YC24 and SZ1 groups compared with the control group (Table 7). At day 42, significant differences in BW were found among B, C, and YC24 groups and the control group. No significant difference was found between the control group and the A and SZ1 groups. 

The determination of BWG indicated that, during weeks 0–3, compared with the control, chicks fed diets supplemented with the combinations of active components had higher weight gain, with a significant difference (*p* < 0.05) observed for A, B, and C combinations and the YC24 and SZ1 groups. Significant decrease in FI was observed in the YC24 group, as well as B and C combinations, compared with the control group whereas, no significant difference was recorded for the A and SZ1 groups. Significantly lower FCR (*p* < 0.05) was observed in YC24 group and A–C combinations compared with the control. No difference in FCR was observed between the control and SZ1 group (*p* > 0.05).

During weeks 4–6, there was no difference in BWG between the control and all supplemental combination diets. A significant decrease (*p* < 0.05) in FI was observed in the YC24 and SZ1 group and A, B, and C combinations compared with the control diet. Moreover, broilers in the A, B, and C combination groups and YC24 or SZ1 supplementation groups had significantly lower FCR (*p* < 0.05) in comparison with broilers in the control treatment.

During weeks 0–6, chicks fed diets supplemented with different combinations had higher BWG compared with those fed the control diets. A statistically significant difference (*p* < 0.05) in BWG was found for broilers receiving the group B-, C-, and YC24-supplemented diets, and there was no significant difference between the A, SZ1, and control groups. In addition, no significant difference in FI was found between the treatment groups. Broilers receiving the group B-, C-, and YC24-supplemented diets had significantly lower (*p* < 0.05) FCR than those fed the control diets. However, no differences were observed between the chicks fed diets supplemented with SZ1, A, and the control diets.

#### 3.4.2. Biochemical Profiles

As shown in Table 8, compared with the control diet, the combination of active components of YC supplements did not affect the TP, ALP, and AST levels, whereas ALT was significantly (*p* < 0.05) decreased in A and C combinations. The BUN level was significantly decreased in broilers fed with A and C combination diets compared with the control group (*p* < 0.05). No difference in BUN level was observed between the SZ1, YC24, and control groups (*p* > 0.05).

#### 3.4.3. Serum Immunoglobulins

The contents of IgA and IgG in serum are presented in Figure 5. On day 21, serum IgA content was increased (*p* < 0.05) in broilers fed with SZ1 compared with the control group. No significant differences were obtained with the combination of active component-supplemented diets as compared with the broilers fed the basal diet. On day 42, combinations A-, B-, C-, and SZ1-supplemented diets significantly increased serum IgA (*p* < 0.05) content. For IgG, significant (*p* < 0.05) differences were found between the control group and the SZ1, A, C, and YC24 groups on day 21. On day 42, the level of IgG was significantly increased in the SZ1, B, C, and YC24 groups (Figure 5).

## 4. Discussion

In the present study, we aimed to investigate the effect of YC obtained at different fermentation time-points on different biological and physiological processes in broilers. The discussion of the results is presented below.

### 4.1. Analysis of YC Components

In a first step, we generated different types of YCs based on the fermentation time and found that the composition profiles of different YCs were influenced by the fermentation time. Especially, YC24 and YC60 were chosen for differential analysis and the results showed that five differentially expressed metabolites could be identified between the YC24 and YC60 groups, namely sucrose, inositol, glycine, fructose, and galactose. The contents observed for glycine, fructose, and inositol corroborated with the content of the YC product SZ1 [12]. In addition, pathway analysis indicated that, except for inositol, the content of other metabolites in the YC24 group was higher than that in the YC60 group. The effectiveness of YC is related to its composition in the effective components. Glycine, as a nutritive additive in broiler feed, is a necessary functional amino acid for poultry; promotes immune regulation by participating in the synthesis of creatine, uric acid, glutathione, and other substances in the body; and is used as an antioxidant and for its other biological functions [12,15,16]. Studies have shown that the addition of glycine to low-protein diets can increase the growth efficiency of broilers, and the growth performance is comparable to that with normal protein diets [17,18,19]. The supplementation of fructose in broiler diets not only improves palatability, but also improves the performance of broilers [20]. Sucrose, as a safe nutritive sweetener, can improve the activity of amino acids and promote the synthesis of proteins by increasing the energetic synthesis [21,22,23]. Galactose is an important component of some glycoproteins that is rapidly absorbed in the intestine and provides energy for the body [24]. Inositol is an important water-soluble vitamin and a biologically active cyclohexanol. Inositol mainly exists in biofilms in the form of phosphatidylinositol and is involved in protein synthesis, lipid metabolism, and the regulation of a series of cell physiological processes. Inositol deficiency in feed can lead to multiple deficiencies such as loss of appetite, anemia, poor growth, fin erosion, skin blackening, and abnormal lipid metabolism in fish [25,26]. In addition, the effect of YC is not the function of a single substance, but the nutrition and health effect of these components in combination [27]. The above findings indicated that the obtained YCs contain important nutritional element and could be used for supplementation in animal feeds or preparation of dietary supplements for animals.

### 4.2. Effect of YCs on Growth Performance of Broilers

To investigate the growth-promoting effect in broilers of different fermentation durations of YC, dietary supplementation was performed. We found that supplementation with the different kinds of YCs had a significant improvement effect on broiler growth and FI during the entire experimental period. This result was corroborated with an earlier study [5] demonstrating that a dietary supplementation of 0.25% YC improves the average daily gain and feed efficiency by about 5% during grower and overall periods. Consistent with this study, the beneficial effects of YC on performance have also been reported for broilers [28] and nursery pigs [29]. Onifade [30] reported that dried yeast (*S. cerevisiae*) given as a supplement to broilers fed with a high fiber diet can improve BWG. At the same time, feed efficiency was also improved in broilers fed with 0.15% and 0.45% of dried yeast. Other studies, however, have reported that yeast products have no effect on the performance of turkey poults [28,29,31], broilers [32,33,34], and broiler breeders [8]. Broilers in the YC24 treatment had higher BWG and improved FCR in comparison with broilers in the control treatment, which indicated that dietary supplementation of YC with a fermentation duration of 24 h had the best performance, while supplementation of YC obtained after a fermentation duration of 60 h had the poorest performance. YC contained yeast cells as well as metabolites such as peptides, organic acids, oligosaccharides, amino acids, flavor and aroma substances, and possibly some unidentifiable growth factors that may have a beneficial effect on animal production. This indicated that YC produced with different fermentation durations contains different metabolites, resulting in different application effects.

To test the combination efficiency of the active metabolites identified, the active ingredients were combined for preparation of dietary supplements and the animal production trials were performed. The results indicated that the combination of active ingredients as dietary additives had reached the application effect of YC24, which was consistent with the results of several previous reports that supplementation of YC to the diet improved the growth performance of broilers [5,33,35,36]. In contrast, while the combination of active ingredients improved the growth performance of broilers, the degree of improvement was not the same, which may be related to the nature and the proportions of nutrients in the different YCs. Indeed, metabolites produced by YC, such as amino acids, small peptides, organic acids, oligosaccharides, flavor enhancers, and aromatic substances, may act synergistically to improve intestinal health and promote digestion, absorption, and utilization of nutrients. This synergistic effect may improve the growth performance and immune function of animals and their overall health. Under our experimental conditions, five main active compounds were identified as potential nutritional additives in the production of broilers. However, the composition of YC is complex, and the roles played by these main active ingredients, as well as the interactions between them, need to be further explored. This is a preliminary study of the functional components of YC, and it needs to be further studied using different research methods and techniques in the future.

### 4.3. Biochemical Profiles

Dietary yeast supplements significantly affect blood parameters, and changes in the serum biochemical parameters can be used to assess both the pathological and nutritional status of an individual. In this study, the serum TP content of broilers was considerably increased by the addition of *S. cerevisiae* YCs obtained at different fermentation times. Similar results in the serum levels of TP, AST, and ALP were previously reported [37]. The study by Shareef and S. A. Al-Dabbagh [38] showed that feed supplemented with 1.5% and 2% *S. cerevisiae* significantly increased the TP and albumin contents in the serum (*p* < 0.05). This implied that the addition of YC in the diet enhanced protein metabolism, absorption, and utilization in broilers and promoted the growth of tissues and organs. However, A Onifade, et al. [39] reported that the serum protein level was decreased and the serum levels of ALT, AST, and ALP were increased with dietary YC in rabbits.

The content of BUN reflects the metabolism of protein in the animal body. A low level of urea nitrogen indicates a good balance in amino acids, increased protein deposition, and normal metabolism of protein. In this study, the BUN content in broilers fed with YC24 and YC36 at day 42 was significantly lower than that of the control group (*p* < 0.05), indicating that *S. cerevisiae* culture can improve the immune function and enhance the metabolism of nutrients, which was consistent with the improved production performance of broilers. In trial 2, the content of BUN in each treatment group was significantly lower than that in the control group (*p* < 0.05), which was similar to findings reported by Zhang, et al. [40]. This may be because the addition of YC promoted the absorption of amino acids and the synthesis of proteins in the animal body and reduced the catabolism of proteins. Serum ALP is closely related to calcium and phosphorus metabolism in immature animals and plays an active role in osteogenesis. Therefore, serum ALP content can indirectly affect the growth of animals. In this paper, the serum ALP content of each experimental group was greater than that of the control group. The results were similar to those reported by Onifade et al. [39].

ALT and AST in the serum are two important enzymes involved in nitrogen metabolism and play an indispensable role in amino acid transformation. The results showed that AST content in the YC group was decreased compared with the control group. This was explainable by the fact that the addition of YC stimulates the animals to feed actively, which often leads to excessive intake. Indeed, in the feeding stage, the broilers are housed in a small area and lack exercise. Consumption of a large amount of feed results in the formation of body fat deposition, and thus weight gain. However, with the accumulation of body fat, excessive nutrients cannot be metabolized smoothly, which damages the liver and leads to the increase of serum aminotransferase content.

### 4.4. Serum Immunoglobulins

To investigate the efficacy of YCs and their active ingredients, we measured their effect on the immune function. Immune factors such as IgG and IgA were increased in the YC24 group compared with the control group. Consistent with this study, Gao et al. [5] showed that the addition of YC can significantly improve broiler serum IgM and sIgA content. The results were also similar to other previous findings [5,40,41]. Our results also showed that the diet supplemented with the combinations of effective components could increase the levels of serum IgG and IgA in broilers to greater levels than that in the YC24 group. Thus, we stipulated that the combination of effective compounds can enhance the immune function and contribute to the health and growth of the broilers. This may be because of the synergistic effect of components in the effective group.

### 4.5. Limitations

Our study presents some limitations. A clear response on DWG, FI, and FC was observed, especially with YC24. However, the “control” group performed quite modestly. Normally, AA birds have an FCR between 0 and 6 w of around 1.60. This may be because of the avidity of broilers for the basal diet or the quality of the basal diet. As a consequence, there may exist a large room for improvements of the YC treatments. Additionally, we observed difference in the FCR of broilers of the same group in both trials. This may be explained by the fact that animal production trials (trials 1 and 2) were performed in the same farm, but in different seasons, which may affect the results.

## 5. Conclusions

In this study, the effective metabolites of YC were screened using metabolomics. The effective compounds of YC were glycine, fructose, inositol, galactose, and sucrose. Moreover, the dietary supplementation of the effective compounds in combination improved BWG, FC, and the immunity of broiler chickens. The results of the present study provide a better understanding of the metabolism of YC, which can help elucidate the nutritional and health care mechanism for the animals in the future. Ultimately, the present study offered an insight into the composition and effect of YC on the growth, biochemical, and immune performance of broilers and provided an idea of the formulation of effective compounds as feed additives for broilers.

## Figures and Tables

**Figure 1 animals-10-00068-f001:**
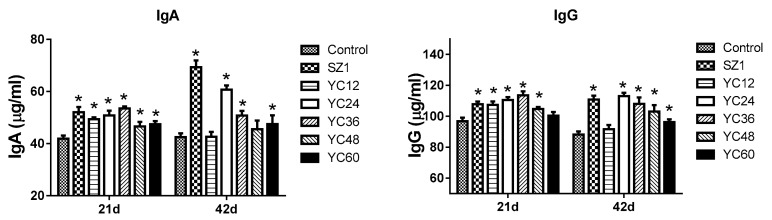
Effects of dietary yeast culture (YC) supplement on the concentrations of immunoglobulins in broiler serum at 21 and 42 days. IgA = immunoglobulin A, IgG = immunoglobulin G. The asterisk (*) denotes significant differences (*p* < 0.05) compared with the control group using one-way analysis of variance (ANOVA). Values are means ± standard deviation.

**Figure 2 animals-10-00068-f002:**
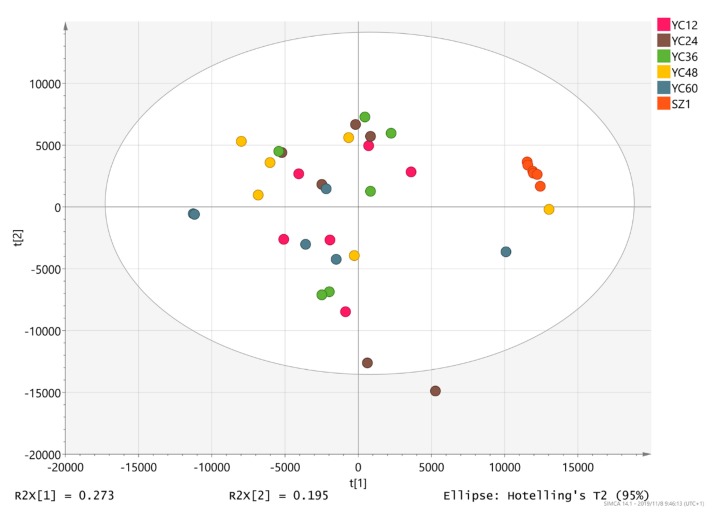
The 2D principal component analysis (PCA) score map.

**Figure 3 animals-10-00068-f003:**
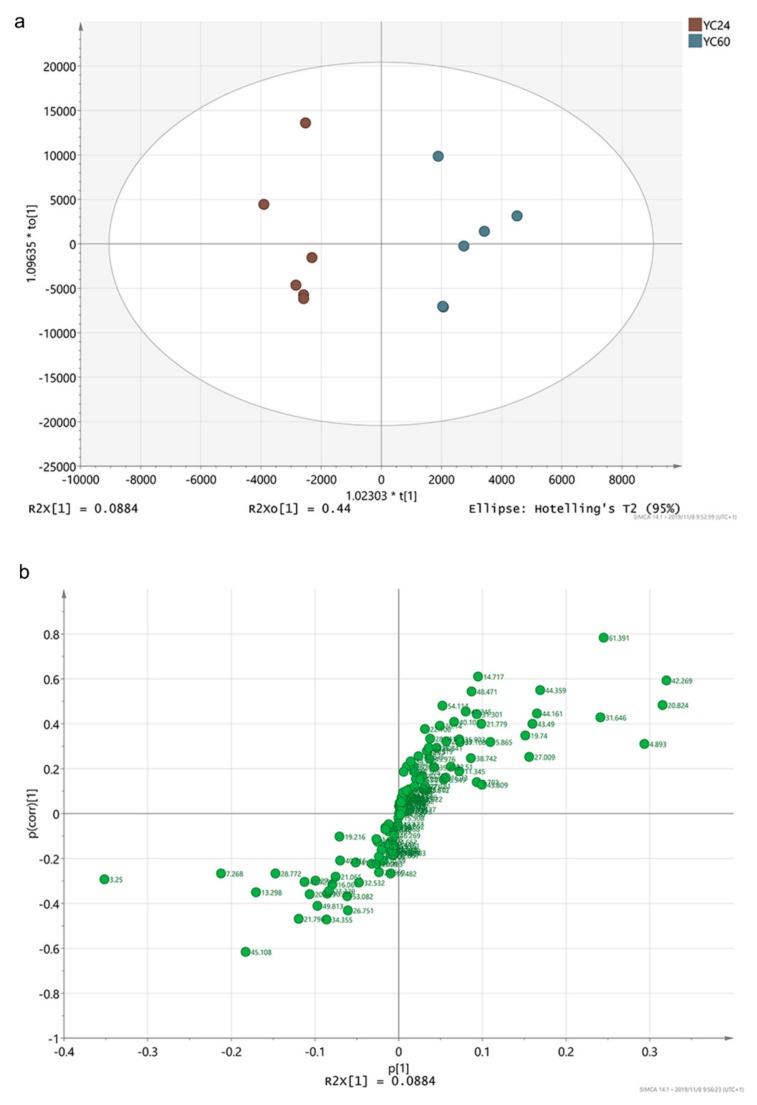
(**a**) Two-dimensional orthogonal partial least squares discriminant analysis (OPLS-DA) score map; (**b**) S-plot.

**Figure 4 animals-10-00068-f004:**
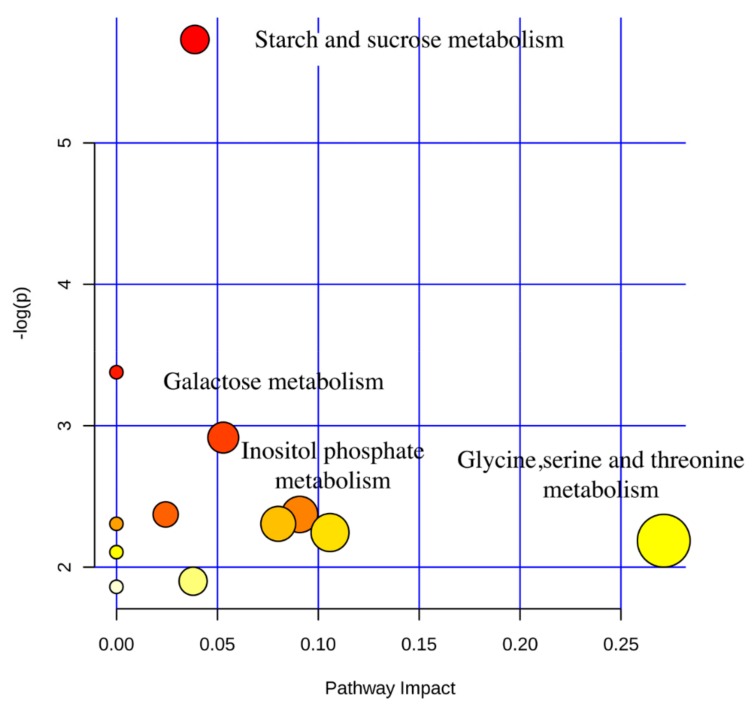
The metabolome view map of the functional impact pathways identified in active components of yeast culture. The x-axis represents the pathway enrichment and the y-axis represents the impact pathway. Larger sizes and darker colors represent the higher pathway enrichment and higher impact pathway values, respectively.

**Figure 5 animals-10-00068-f005:**
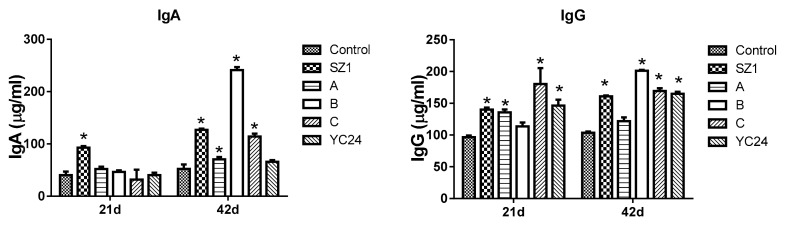
Effect of dietary combination of active components of YC supplement on the concentrations of immunoglobulins in broiler serum at 21 and 42 days. The asterisk (*) denotes significant differences (*p* < 0.05) compared with the control group, as analyzed by one-way analysis of variance (ANOVA) followed by Bonferroni multiple comparison tests.

**Table 1 animals-10-00068-t001:** Ingredients and nutrient levels of the basal starter (1–21 days) and finisher (22–42 days) diets (on dry matter basis) used in experiments 1 and 2.

Ingredient (g/kg)	Starter (1–21 Days)	Finisher (22–42 Days)
Corn	52.07	55.57
Soybean meal	35.50	34.00
Fish meal	5.00	3.60
Soybean oil	4.00	3.50
phosphate	1.00	0.70
Limestone	1.00	1.20
Salt	0.30	0.30
DL-methionine	0.09	0.09
1% premix ^a^	1.0	1.0
Total	100	100
Nutrition value		
Metabolizable Energy (ME) MJ/kg	12.75	12.72
Protein %	20.72	19.61
Calcium %	0.91	0.87
Total phosphorus %	0.68	0.58
Available phosphorus	0.45	0.36
Lysine %	1.35	1.25
Methionine %	0.46	0.43

^a^ 1% premix provides the following per kilogram of feed: vitamin A, 5000 international unit (IU); vitamin D, 1500 IU as vitamin D3; vitamin E, 15 IU; vitamin B12, 0.01 mg; vitamin K3, 0.8 mg; folic acid, 0.5 mg; nicotinic acid, 50 mg; pantothenic acid, 8 mg; biotin, 0.1 mg; pyridoxine, 2.2 mg; riboflavin, 4.4 mg; and thiamine mononitrate, 1.6 mg. Mineral premix contained per kilogram of diet: Fe, 80 g; Cu, 6 mg; Mn, 100 mg; Zn, 80 mg; I, 0.4 mg; and Se, 0.2 mg.

**Table 2 animals-10-00068-t002:** Identification of metabolites of yeast culture.

Name		Relative Peak Area of Fermented Product					
		YC12	YC24	YC36	YC48	YC60	SZ1
1	Styrene	N/A	N/A	N/A	N/A	N/A	4.4 × 10^6^
2	Propanoic acid	1.1 × 10^9^	1.3 × 10^9^	1.1 × 10^9^	1.3 × 10^9^	1.7 × 10^9^	5015342
3	Mercaptoacetic acid	N/A	N/A	N/A	N/A	5.8 × 10^7^	N/A
4	Acetic acid	7.1 × 10^7^	8.7 × 10^7^	5.9 × 10^7^	7.7 × 10^7^	6.6 × 10^7^	6.9 × 10^6^
5	L-Valine	2.2 × 10^7^	7.4 × 10^7^	8.5 × 10^7^	1.3 × 10^8^	3.7 × 10^7^	1.8 × 10^7^
6	Phenylpropanolamine	N/A	N/A	N/A	N/A	N/A	1.8 × 10^7^
7	Ethanimidic acid	4.4 × 10^6^	N/A	N/A	1.0 × 10^6^	1.0 × 10^6^	N/A
8	L-(+)-Lactic acid	1.7 × 10^8^	2.1 × 10^8^	1.6 × 10^8^	2.0 × 10^8^	1.9 × 10^8^	2.1 × 10^9^
9	l-Alanine	3.8 × 10^7^	9.7 × 10^7^	1.3 × 10^8^	1.9 × 10^8^	7.4 × 10^7^	1.7 × 10^6^
10	l-Proline	N/A	1.0 × 10^6^	2.2 × 10^7^	N/A	N/A	N/A
11	Propanedioic acid	1.3 × 10^7^	1.0 × 10^6^	N/A	1.0 × 10^6^	1.1 × 10^7^	6.4 × 10^6^
12	Ethyl phosphoric acid	N/A	1.0 × 10^6^	1.0 × 10^6^	1.7 × 10^7^	N/A	2.8 × 10^6^
13	Urea	2.0 × 10^9^	2.8 × 10^9^	9.1 × 10^8^	2.1 × 10^9^	2.3 × 10^9^	1.9 × 10^6^
14	L-Leucine	5.3 × 10^8^	6.5 × 10^8^	3.5 × 10^8^	5.0 × 10^8^	6.7 × 10^8^	3.2 × 10^7^
15	l-Threonine	N/A	1.0 × 10^6^	N/A	1.0 × 10^6^	1.0 × 10^6^	N/A
16	Butane	N/A	N/A	N/A	N/A	N/A	4.1 × 10^6^
17	Glycine	3.6 × 10^8^	1.3 × 10^9^	9.9 × 10^8^	1.4 × 10^9^	1.4 × 10^9^	8.2 × 10^6^
18	Butanedioic acid	1.5 × 10^8^	2.0 × 10^8^	1.7 × 10^8^	2.0 × 10^8^	3.0 × 10^8^	6.8 × 10^7^
19	Glycerol	N/A	N/A	N/A	N/A	N/A	2.2 × 10^9^
20	Pyrimidine	1.2 × 10^8^	1.8 × 10^8^	1.2 × 10^8^	1.9 × 10^8^	2.0 × 10^8^	N/A
21	2-Butenedioic acid	1.6 × 10^7^	2.7 × 10^7^	1.8 × 10^7^	2.2 × 10^7^	1.0 × 10^6^	4.8 × 10^7^
22	Serine	1.5 × 10^8^	2.8 × 10^8^	9.9 × 10^7^	2.5 × 10^8^	1.0 × 10^6^	1.4 × 10^7^
23	l-Methionine	1.0 × 10^6^	1.0 × 10^6^	7.3 × 10^7^	1.0 × 10^6^	1.0 × 10^6^	N/A
24	beta-Alanine	1.0 × 10^6^	1.0 × 10^6^	1.6 × 10^7^	N/A	N/A	N/A
25	ketobutyrate	N/A	N/A	N/A	N/A	6.3 × 10^7^	N/A
26	2-Hydroxycyclohexane-1-carboxylic acid	4.01 × 10^7^	1.03 × 10^6^	1.0 × 10^6^	6.6 × 10^7^	N/A	N/A
27	L-Homoserine	3.00 × 10^7^	N/A	1.9 × 10^7^	3.8 × 10^7^	N/A	N/A
28	Aminomalonic acid	7.69 × 10^7^	7.59 × 10^7^	6.5 × 10^7^	6.8 × 10^7^	9.8 × 10^7^	N/A
29	oxalate	N/A	N/A	N/A	N/A	N/A	3.0 × 10^6^
30	L-Proline	7.2 × 10^8^	1.0 × 10^9^	7.3 × 10^8^	1.2 × 10^9^	1.3 × 10^9^	7.9 × 10^7^
31	L-Aspartic acid	1.3 × 10^8^	2.2 × 10^8^	5.8 × 10^7^	1.2 × 10^8^	1.8 × 10^8^	2.3 × 10^7^
32	Naphthalene	N/A	N/A	N/A	N/A	N/A	4.0 × 107
33	Alanine	8.4 × 10^7^	1.1 × 10^8^	1.5 × 10^8^	1.5 × 10^8^	9.5 × 10^7^	N/A
34	2-Furancarboxylic acid	N/A	1.0 × 10^6^	1.2 × 10^7^	2.2 × 10^7^	2.2 × 10^7^	N/A
35	L-Threonic acid	1.3 × 10^7^	5.6 × 10^7^	2.8 × 10^7^	5.0 × 10^7^	5.0 × 10^7^	3.2 × 10^7^
36	2-Pentenoic acid	1.0 × 10^6^	N/A	9.0 × 10^6^	1.0 × 10^6^	N/A	N/A
37	Xylonic acid	N/A	N/A	N/A	N/A	N/A	2.2 × 10^6^
38	L-Sorbose	N/A	N/A	N/A	N/A	N/A	1.4 × 10^7^
39	Carbamoylglycine	9.9 × 10^7^	1.0 × 10^6^	4.7 × 10^7^	1.1 × 10^8^	1.1 × 10^8^	N/A
40	Ornithine	N/A	1.7 × 10^8^	N/A	7.0 × 10^7^	N/A	N/A
41	Glutamic acid	N/A	N/A	N/A	N/A	N/A	6.4 × 10^6^
42	Xylitol	N/A	N/A	N/A	N/A	N/A	1.3 × 10^7^
43	Tricarballylic acid	N/A	N/A	N/A	N/A	N/A	3.4 × 10^7^
44	Adonitol	N/A	N/A	N/A	N/A	N/A	N/A
45	D-Arabino-Hexonic acid	N/A	N/A	N/A	N/A	N/A	N/A
46	L-Asparagine	2.3 × 10^7^	4.3 × 10^7^	1.5 × 10^7^	4.2 × 10^7^	4.7 × 10^7^	N/A
47	D-(+)-Talose	N/A	N/A	N/A	N/A	N/A	N/A
48	D-(+)-Arabitol	4.3 × 10^7^	3.9 × 10^7^	5.2 × 10^7^	8.1 × 10^7^	8.5 × 10^7^	N/A
49	D-Mannitol	N/A	N/A	N/A	N/A	N/A	N/A
50	Dulcitol	N/A	N/A	N/A	N/A	N/A	N/A
51	1-Propene-1,2,3-tricarboxylic acid	3.3 × 10^7^	4.0 × 10^7^	2.8 × 10^7^	4.2 × 10^7^	3.8 × 10^7^	N/A
52	Benzoic acid	1.6 × 10^7^	1.0 × 10^6^	1.4 × 10^7^	1.9 × 10^7^	1.4 × 10^7^	N/A
53	Phosphoric acid	2.0 × 10^8^	2.7 × 10^8^	2.5 × 10^8^	3.5 × 10^8^	2.4 × 10^8^	N/A
54	Glycylglycine	1.0 × 10^6^	N/A	N/A	N/A	N/A	N/A
55	Ribonic acid	N/A	N/A	N/A	N/A	N/A	9.7 × 10^6^
56	2-Keto-l-gluconic acid	N/A	N/A	N/A	N/A	1.0 × 10^6^	N/A
57	DL-Ornithine	2.5 × 10^7^	8.0 × 10^7^	3.6 × 10^7^	7.6 × 10^7^	7.2 × 10^7^	N/A
58	1,2,3-Propanetricarboxylic acid	2.4 × 10^8^	3.6 × 10^8^	N/A	1.0 × 10^6^	1.0 × 10^6^	3.1 × 10^7^
59	l-Fucitol	4.4 × 10^7^	5.4 × 10^7^	4.3 × 10^7^	5.8 × 10^7^	7.9 × 10^7^	N/A
60	d-Pinitol	3.4 × 10^8^	4.2 × 10^8^	2.9 × 10^8^	4.3 × 10^8^	2.8 × 10^8^	N/A
61	d-(+)-Cellobiose	N/A	N/A	N/A	N/A	N/A	1.0 × 10^7^
62	fructose	1.6 × 10^8^	2.2 × 10^8^	1.9 × 10^8^	2.8 × 10^8^	2.5 × 10^8^	4.3 × 10^7^
63	d-Mannose	N/A	1.0 × 10^6^	N/A	N/A	1.5 × 10^8^	N/A
64	l-Tyrosine	3.1 × 10^8^	4.4 × 10^8^	3.0 × 10^8^	4.7 × 10^8^	6.4 × 10^8^	N/A
65	l-Lysine	N/A	7.6 × 10^8^	N/A	N/A	N/A	N/A
66	d-(−)-Ribose	N/A	N/A	1.0 × 10^6^	1.7 × 10^8^	N/A	N/A
67	Phenol	1.0 × 10^6^	1.0 × 10^6^	1.8 × 10^8^	3.3 × 10^8^	3.9 × 10^8^	N/A
68	Inositol	5.8 × 10^7^	8.6 × 10^7^	6.4 × 10^7^	4.8 × 10^8^	3.7 × 10^8^	6.7 × 10^6^
69	Hexanoic acid	N/A	N/A	N/A	1.0 × 10^6^	N/A	N/A
70	d-(+)-Galactopyranose	7.3 × 10^7^	1.0 × 10^8^	8.3 × 10^7^	1.5 × 10^8^	1.0 × 10^6^	N/A
71	d-Gluconic acid	1.8 × 10^8^	2.5 × 10^8^	2.9 × 10^7^	2.6 × 10^8^	1.4 × 10^8^	4.3 × 10^7^
72	Hexadecanoic acid	1.5 × 10^8^	1.5 × 10^8^	1.2 × 10^8^	2.0 × 10^8^	2.6 × 10^8^	5.2 × 10^8^
73	(Z)-4-Nitro-alpha-(p-nitrophenyl)cinnamic acid	N/A	N/A	N/A	1.5 × 10^8^	N/A	N/A
74	alpha-d-(+)-Talopyranose	2.4 × 10^7^	4.0 × 10^7^	N/A	1.0 × 10^6^	1.3 × 10^8^	N/A
75	l-Tryptophan	N/A	7.8 × 10^7^	1.0 × 10^6^	N/A	N/A	N/A
76	Octadecanoic acid	1.7 × 10^8^	1.6 × 10^8^	2.0 × 10^8^	2.9 × 10^8^	3.3 × 10^8^	N/A
77	Pentanoic acid	N/A	N/A	N/A	1.0 × 10^6^	N/A	N/A
78	Arachidonic acid	N/A	N/A	1.0 × 10^6^	1.8 × 10^7^	1.9 × 10^7^	N/A
79	Morphinan	N/A	1.5 × 10^7^	N/A	N/A	1.0 × 10^6^	N/A
80	1,5-Anhydro-d-sorbitol	7.2 × 10^6^	1.4 × 10^7^	5.0 × 10^7^	1.0 × 10^6^	1.2 × 10^7^	N/A
81	Lactulose	5.9 × 10^6^	N/A	2.9 × 10^7^	N/A	N/A	N/A
82	Decanedioic acid	N/A	N/A	5.4 × 10^7^	N/A	1.0 × 10^6^	N/A
83	Sucrose	3.0 × 10^7^	3.8 × 10^7^	2.5 × 10^7^	N/A	2.7 × 10^8^	6.4 × 10^6^
84	Maltose	3.8 × 10^7^	1.1 × 10^8^	5.7 × 10^7^	1.0 × 10^6^	3.0 × 10^7^	N/A
85	Palatinose	4.3 × 10^7^	3.0 × 10^7^	N/A	N/A	N/A	N/A

N/A indicates that the substance does not exist. YC, yeast culture.

**Table 3 animals-10-00068-t003:** Growth performance of broilers fed diets containing different YCs from week 0 to week 6.

	Experimental Diets							SEM	*p* Value
	Control	YC12	YC24	YC36	YC 48	YC 60	SZ1		
Initial BW (g)	45	45	45	45	45	44	44	0.05	0.06
Day 21 BW (g)	612 ^a^	699 ^b,c^	731 ^c^	718 ^b,c^	663 ^a,b^	632 ^a^	676 ^a,b,c^	9.21	0.00
Day 42 BW (g)	2017 ^a^	2167 ^b^	2433 ^c^	2319 ^c^	2162 ^b^	2086 ^a,b^	2166 ^b^	26.63	0.00
0–3 week									
BWG (g)	567 ^a^	654 ^a,b,c^	686 ^b,c^	674 ^a,b,c^	619 ^a,b^	588 ^a^	628 ^a,b,c^	9.20	0.00
FI (g)	1032 ^a^	1116 ^a,b^	1081 ^a,b^	1138 ^b^	1084 ^a,b^	1069 ^a,b^	1082 ^a,b^	9.47	0.08
FCR	1.82 ^a^	1.7 ^a,b^	1.57 ^b^	1.69 ^a,b^	1.74 ^a^	1.81 ^a^	1.73 ^a^	0.02	0.03
4–6 week									
BWG (g)	1398 ^a,b^	1467 ^c,d^	1691 ^e^	1503 ^d^	1411 ^a,b,c^	1361 ^a^	1448 ^b,c,d^	17.02	0.00
FI (g)	2452 ^a^	2488 ^a,b^	2588 ^b,c^	2483 ^a,b^	2643 ^c^	2570 ^a,b,c^	2470 ^a,b^	17.82	0.02
FCR	1.75 ^a^	1.69 ^a,b^	1.53 ^c^	1.65 ^b^	1.87 ^d^	1.89 ^d^	1.7 ^a,b^	0.02	0.00
0–6 week									
BWG (g)	1972 ^a^	2123 ^b^	2388 ^c^	2275 ^c^	2118 ^b^	2043 ^a,b^	2122 ^b^	25.48	0.00
FI (g)	3484 ^a^	3610 ^a,b^	3687 ^b,c,d^	3730 ^b,c,d^	3835 ^d^	3813 ^c,d^	3667 ^b,c^	24.99	0.00
FCR	1.76 ^a,b,c^	1.7 ^a,b^	1.54 ^d^	1.64 ^c,d^	1.81 ^d,e^	1.86 ^e^	1.72 ^c,d^	0.02	0.00

BW = body weight; BWG = body weight gain (g/w); FI = feed intake (g/w); FCR = feed conversion ratio (g/g), SEM= standard error of the mean. ^a–e^ means within the same rows without the same superscript letter are significantly different (*p* < 0.05).

**Table 4 animals-10-00068-t004:** Effects of dietary fermentation time of yeast culture on some plasma biochemical parameters of broilers over 42 days.

	Experimental Diets							SEM	*p* Value
	Control	YC12	YC24	YC36	YC 48	YC 60	SZ1		
TP (g/L)	30.75 ^a^	31.64 ^b^	31 ^a,b^	32.66 ^c^	34.9 ^d^	30.14 ^a^	33.86 ^e^	0.26	0.00
ALP (U/L)	1675.4 ^a^	1905.91 ^b^	2064.67 ^d^	1974.53 ^b^	1886.64 ^c^	1875.6 ^c^	1764.54 ^a,c^	18.56	0.00
AST (U/L)	279.33	318.78	276.72	303.31	280.6	273.38	302.52	6.38	0.51
ALT (U/L)	47.91 ^a^	48.51 ^a,b^	52.24 ^c^	52.69 ^c^	53.73 ^d^	56.02 ^e^	49.29 ^b^	0.44	0.00
BUN (m mol/L)	0.73 ^a^	0.72 ^a^	0.52 ^b^	0.59 ^b,c^	0.63 ^c^	0.62 ^c^	0.6 ^c^	0.01	0.00

TP = total protein; ALT = alanine aminotransferase; ALP = alkaline phosphatase; AST = aspartate aminotransferase; BUN = blood urea nitrogen. ^a–e^ means within the same rows without the same superscript letter are significantly different (*p* < 0.005).

**Table 5 animals-10-00068-t005:** Biomarkers of yeast cultures. VIP = very important parameter.

Name	VIP	*p* Value
Glycine	3.9	0.049
fructose	4.7	0.041
Inositol	1.7	0.048
D-(+)-Galactopyranose	2.1	0.049
sucrose	3.4	0.002

**Table 6 animals-10-00068-t006:** Combination schemes of the active component for the preparation of feed additives. According to the regression model, the production performance was obtained as the daily body weight gain with supplementation of yeast culture fermented for 24 h. The corresponding value was obtained for each active component. On this basis, the proportion of each component was adjusted (each component + 10%) to configure the content of each component for groups A–C.

	The Proportion of Substances in YC %				
Combination	Glycine	Fructose	Inositol	Galactose	Sucrose
Group A	4.1	0.6	0.5	0.8	0.1
Group B	3.7	0.6	0.5	0.8	0.1
Group C	4.5	0.7	0.6	0.9	0.1

**Table 7 animals-10-00068-t007:** The effect of different treatment of yeast culture on the growth performance of broilers. BW = body weight; BWG = body weight gain (g/w); FI = feed intake (g/w); FCR = feed conversion ratio (g/g). ^a–c^ means within the same rows without the same superscript letter are significantly different (*p* < 0.05).

	Experimental Diets						SEM	*p* Value
	Control	A	B	C	YC 24	SZ1		
Initial BW (g)	44.47	44.41	44.53	44.57	44.63	44.14	0.08	0.58
Day 21 BW (g)	756.95 ^a^	845.08 ^b^	865.4 ^b^	853.3 ^b^	832.29 ^b^	838.63 ^b^	10.88	0.03
Day 42 BW (g)	2091.18 ^a^	2221.49 ^a^^,b,c^	2280.8 ^b,c^	2375.6 ^c^	2330.78 ^b,c^	2189.04 ^a,b^	26.69	0.01
0–3 week								
BWG (g)	712.47 ^a^	800.67 ^b^	820.86 ^b^	808.73 ^b^	787.67 ^b^	794.48 ^b^	10.88	0.03
FI (g)	1229.83 ^a,b^	1205.3 ^a^	1082.27 ^c^	1034.47 ^d^	1049.67 ^d,c^	1257.13 ^b^	17.01	0.00
FCR	1.74 ^a^	1.55 ^b^	1.35 ^c^	1.31 ^c^	1.37 ^c^	1.63 ^a,b^	0.03	0.00
4–6 week								
BWG (g)	1272.27	1280.46	1316.31	1331.66	1380.01	1257.75	16.45	0.27
FI (g)	2897.12 ^a^	2341.33 ^b^	2529.74 ^c^	2495.43 ^c^	2561.18 ^c^	2297.15 ^b^	38.88	0.00
FCR	2.28 ^a^	1.83 ^b^	1.93 ^b^	1.88 ^b^	1.86 ^b^	1.82 ^b^	0.03	0.00
0–6 week								
BWG (g)	2046.7 ^a,b^	2177.08 ^a,b,c^	2236.27 ^b,c^	2331.03 ^c^	2286.15 ^b,c^	2144.89 ^a,b^	26.66	0.01
FI (g)	3676.24	3588.21	3572.36	3626.69	3578.06	3611.59	22.62	0.79
FCR	1.76 ^a^	1.65 ^a,b^	1.6 ^b^	1.55 ^b^	1.56 ^b^	1.68 ^a,b^	0.02	0.02

**Table 8 animals-10-00068-t008:** Effects of different dietary treatments on some plasma biochemical parameters of broilers over 42 days.

	Experimental Diets							
	Control	A	B	C	YC 24	SZ1	SEM	*p* Value
TP (g/L)	31.68	30.26	32.05	32.16	31.12	31.47	0.33	0.62
ALP (U/L)	3751.62	2732.07	3781.78	4193.26	2435.39	3034.93	274.77	0.45
AST (U/L)	285.17	302.93	318.83	276.76	303.3	273.42	7.38	0.41
ALT (U/L)	50.89 ^a^	48.7 ^b^	49.82 ^a,b^	48.77 ^b^	50.42 ^a^	49.59 ^a,b^	0.22	0.02
BUN (mmol/L)	0.7 ^a,c^	0.53 ^b^	0.71 ^c^	0.52 ^b^	0.59 ^a,b^	0.54 ^b^	0.01	0.001

^a–c^ means within the same rows without the same superscript letter are significantly different (*p* < 0.005).

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
