# Peer review of "Effect of Yeast Culture (Saccharomyces cerevisiae) on Broilers: A Preliminary Study on the Effective Components of Yeast Culture"

_animals, 2019, doi:10.3390/ani10010068_

Round 1

Reviewer 1 Report

The introduction and the material and methods sections have been improved. Results have been reanalyzed as requested. However, the results section description is still unclear and misleading. Unless these flaws are accounted for I cannot find this manuscript acceptable for publication.

Major concerns

Line 206. You performed a Bonferroni multiple comparison test. That is ok. But do you represent in tables a multiple comparison test or just a comparison against the control? It is unclear and misleading. In addition, if authors just performed comparisons against the control, first, I do not see what the use of treatment SNZ1 is, and second, there is no clear evidence to select YC time to perform the second trial.

Tables are unclear and misleading. Productive tables: could you please provide final body weight?

Other concerns

Line 44-49. Please delete. You have enough evidence with poultry.

Line 73 and 75: molelcules instead of variables.

Line 113 and elsewhere, try to use the same units along the manuscript: either “g” or “rpm”

Line 162. Animal production trials instead of animal experiments

Line 172. Check subheading it is the same as line 164.

Line 183. Group D does not appear anymore. Please either delete or reword.

Table 3:

BWG (0-3 weeks): You provide superscripts "a" for YC12, 24 and 36, compared to what? You need to provide superscripts for Control, SNZ1, YC48 and 60.

BWG (4-6 weeks): All treatment groups with the same superscript? In table caption it is stated "b denotes significant differences compared to 0-3 weeks" This is very much misleading. In addition, this has not been described in the statistical analysis of the material and methods section.

FI (4-6 weeks): You provide superscripts "b" and "ab". You cannot have a superscript "ab" without a superscript "a", and you do not have a superscript "a". Please check statistics outputs.

BWG (0-6 weeks): You provide superscripts "b" and "ab". You cannot have a superscript "ab" without a superscript "a", and you do not have a superscript "a". Please check statistics outputs.

FI (0-6 weeks): You provide superscripts "b" and "ab". You cannot have a superscript "ab" without a superscript "a", and you do not have a superscript "a". Please check statistics outputs.

FCR (0-6 weeks): You provide superscripts "a" for YC24 compared to what? You need to provide superscripts for all other treatments.

What is the use of providing SD and SEM both at the same time? Use one or the other. Provide the true P value in one additional. Provide no decimals for BWG or FI and 2 decimals for FCR. You find very low SD or SEM for FCR. This implies that you will find narrow differences as significant, but are these significant differences relevant from a productive point of view? This should be addressed in the discussion. Define all the abbreviations used in the table.

Line 219-221. You are performing an "all by all" comparison, please describe whether YC12, 24 or 36 are different from any other treatment. This is important because you are going to choose one specific fermentation time.

Line 223-225. Not true according to superscripts in table 3. For that to be true superscript for YC36 should have been "a" not "ab". Again, please check superscript assignment. Describe the all by all comparison.

Line 233-235, is this table heading? Provide units for the measured variables.

Line 237-246 Authors found significant differences but should state whether they found an increase or a decrease.

Table 4. Provide superscripts for the all by all comparison. For instance, superscripts for control, YC12 and 24 in TP analysis. 

Line 248-250. Is this table heading? it should be table caption.

Line 263-265. This is not explained in the statistics section. Please either delete or describe.

Is it not Figure 2 the same as Figure 3a?

Line 281, on what grounds? Which YC has the greatest content in effective metabolites? Or are there any differences between YCs?

Line 282-284 is not this information repeated in Line 274-278?

Is figure 3b not used?

Table 5. Define all abbreviations used in table. Provide 3 decimals for P value. One of your selection criterion is that P<0.05, and in Table 5 there are several biomarkers with P=0.05. Is it necessary to provide Rt min?

Line 309-312. It is not clear at all that YC24 significantly differed from any other YC. This has to be clearly presented in tables and described in the text in order to justify this selection.

Table 6. What is the meaning of X1…X6? Is it necessary?

Line 332-333. Define whether it increased or decreased.

Line 333-334. You found a significant decrease in FCR. Is a FCR of 2.3 in the control group usual? In trial 1 you found a FCR of 1.7. Did you change concentrate formulation for trial 2?

Line 334-336. Not true, just C as you state in the following sentence. Please reword.

Line 337-338. Not true. Please delete.

Line 340-341. Delete.

Line 343-345. This is table caption, not heading

Table 7. The same as for table 3.

Line 357. This is table caption

Line 364-365 and SZ1

Line 367-368. Please delete.

Author Response

Major concerns

Line 206. You performed a Bonferroni multiple comparison test. That is ok. But do you represent in tables a multiple comparison test or just a comparison against the control? It is unclear and misleading. In addition, if authors just performed comparisons against the control, first, I do not see what the use of treatment SNZ1 is, and second, there is no clear evidence to select YC time to perform the second trial.

Response: the data have been reanalyzed as described below: “The data were compared in terms of means, pooled SEM, and p-values using a one-way ANOVA model, significant difference between groups were analyzed by Duncan's multiple comparisons, and significance was declared at p < 0.05. Data were also analyzed by SPSS 23.0 for correlation between the active ingredient content of the selected YC and BWG, and the stepwise regression method was used to establish the prediction equation which was used for prediction the proportion of key metabolites in combination diets.”

SZ1 is a commercial yeast culture product, which is similar to our fermented products, so it was selected as a positive control for comparison of feeding effects.

The selection of the fermentation time of YC is mainly based on the results of production performance, taking into account indicators such as immune function, and considering production costs. Comprehensive evaluation results show that the best effect of fermentation is 24 hours.

Tables are unclear and misleading. Productive tables: could you please provide final body weight?

Response: the data have been reorganized and the tables improved. The initial body weight and body weight at day 21 and day 42  

Other concerns

Line 44-49. Please delete. You have enough evidence with poultry.

Response: This section has been deleted as recommended

Line 73 and 75: molelcules instead of variables.

Response: “variables”  has been changed to  molecules where necessary as recommended.

Line 113 and elsewhere, try to use the same units along the manuscript: either “g” or “rpm”.

Response: the units have been uniformized as recommended.

Line 162. Animal production trials instead of animal experiments

Response: “animal experiments” has been changed to “animal trials”

Line 172. Check subheading it is the same as line 164.

Response: we have corrected the subheading as recommended.

Line 183. Group D does not appear anymore. Please either delete or reword.

Response: we have deleted “group D” and rephrased appropriately.

Table 3:

BWG (0-3 weeks): You provide superscripts "a" for YC12, 24 and 36, compared to what? You need to provide superscripts for Control, SNZ1, YC48 and 60. BWG (4-6 weeks): All treatment groups with the same superscript? In table caption it is stated "b denotes significant differences compared to 0-3 weeks" This is very much misleading. In addition, this has not been described in the statistical analysis of the material and methods section. FI (4-6 weeks): You provide superscripts "b" and "ab". You cannot have a superscript "ab" without a superscript "a", and you do not have a superscript "a". Please check statistics outputs. BWG (0-6 weeks): You provide superscripts "b" and "ab". You cannot have a superscript "ab" without a superscript "a", and you do not have a superscript "a". Please check statistics outputs. FI (0-6 weeks): You provide superscripts "b" and "ab". You cannot have a superscript "ab" without a superscript "a", and you do not have a superscript "a". Please check statistics outputs. FCR (0-6 weeks): You provide superscripts "a" for YC24 compared to what? You need to provide superscripts for all other treatments. What is the use of providing SD and SEM both at the same time? Use one or the other. Provide the true P value in one additional. Provide no decimals for BWG or FI and 2 decimals for FCR.

Response: Table 3 has been revised and the new presentation contains the pooled SEM and the p-values for the comparisons. Superscript letters have been used for indicating similar group and groups presenting significant differences.

You find very low SD or SEM for FCR. This implies that you will find narrow differences as significant, but are these significant differences relevant from a productive point of view? This should be addressed in the discussion.

Define all the abbreviations used in the table.

Response: the abbreviations used in the tables have been defined in the text and in the table caption.

Line 219-221. You are performing an "all by all" comparison, please describe whether YC12, 24 or 36 are different from any other treatment. This is important because you are going to choose one specific fermentation time.

Response: differences and similarities between the treatment groups have been presented using superscript letter in the tables, and the results have been described appropriately in the text.

Line 223-225. Not true according to superscripts in table 3. For that to be true superscript for YC36 should have been "a" not "ab". Again, please check superscript assignment. Describe the all by all comparison.

Response: differences and similarities between the treatment groups have been presented using superscript letter in the tables, and the results have been described appropriately in the text.

Line 233-235, is this table heading? Provide units for the measured variables.

Response: the table caption has been presented appropriately and the units of variables provided accordingly.

Line 237-246 Authors found significant differences but should state whether they found an increase or a decrease.

Response: we have specified whether the changes were an increase or decrease throughout the manuscript.

Table 4. Provide superscripts for the all by all comparison. For instance, superscripts for control, YC12 and 24 in TP analysis. 

Response: differences and similarities between the treatment groups have been presented using superscript letter in the tables, and the results have been described appropriately in the text.

Line 248-250. Is this table heading? it should be table caption.

Response: the table caption has been presented appropriately.

Line 263-265. This is not explained in the statistics section. Please either delete or describe.

Response: we have deleted this and revised the figures appropriately.

Is it not Figure 2 the same as Figure 3a?

Response: the referencing for Figure 2 and figure 3a are correct. We have revised the text for better understanding.

Line 281, on what grounds? Which YC has the greatest content in effective metabolites? Or are there any differences between YCs?

Response: this sentence has been deleted. Based on the production performance and the biochemical and immunological indices, YCs were considered to contain effective metabolites when they had exhibited good performance. Since YC60 showed the worst production performance and YC24 exhibited the best production performance, these two YCs were used for screening differential effective metabolites. This has been stated in the revised manuscript.

Line 282-284 is not this information repeated in Line 274-278?

Response: we are sorry for the mistake. Figure 3a has been changed to the correct one and the data described accordingly

Is figure 3b not used?

Response: Figure 3b has been described and referenced as recommended and as follows: “The S-plot showing the key metabolites was depicted in Figure 3b. On the left and right side of the S-plot, the variables with strong model contribution and high statistical reliability were screened for uncovering potential biomarkers to characterize the metabolic discriminations between YC24 and YC60.”

Table 5. Define all abbreviations used in table. Provide 3 decimals for P value. One of your selection criterion is that P<0.05, and in Table 5 there are several biomarkers with P=0.05. Is it necessary to provide Rt min?

Response: Rt min has been deleted and the abbreviations defined. Actually the metabolites were screened at p<0.05. the p value have written with 3 digit and the table revised.

Line 309-312. It is not clear at all that YC24 significantly differed from any other YC. This has to be clearly presented in tables and described in the text in order to justify this selection.

Response: Based on the production performance and the biochemical and immunological indices, YCs were considered to contain effective metabolites when they had exhibited good performance. Since YC60 showed the worst production performance and YC24 exhibited the best production performance, these two YCs were used for screening differential effective metabolites.

Table 6. What is the meaning of X1…X6? Is it necessary?

 Response: X1-X5 have been deleted

Line 332-333. Define whether it increased or decreased.

Response: we have specified that it was “decreased” in the revised manuscript.

Line 333-334. You found a significant decrease in FCR. Is a FCR of 2.3 in the control group usual? In trial 1 you found a FCR of 1.7. Did you change concentrate formulation for trial 2?

Response: animal production trials (trial 1 and trial 2) were performed on the same farm, but in different seasons, this may affect the results. We have mentioned this in the limitations.

Line 334-336. Not true, just C as you state in the following sentence. Please reword.

Response: the data have reanalyzed and described accordingly

Line 337-338. Not true. Please delete.

Response: the data have reanalyzed and described accordingly

Line 340-341. Delete.

Response: the data have reanalyzed and described accordingly

Line 343-345. This is table caption, not heading

Response: the table caption has been revised appropriately

Table 7. The same as for table 3.

Response: Table 7 has been revised accordingly.

Line 357. This is table caption

Response: the Table caption has been presented appropriately.

Line 364-365 and SZ1

Response: SZ1 has been added to the groups with significant effect on IgA

Line 367-368. Please delete.

Response: This sentence has been deleted accordingly.

Reviewer 2 Report

The authors have addressed all concerns.

Just a minor with references

Reference (1) is missing and  NOT indicated in the text (Reference numbering starts with 2).

Also Reference (1) listed in the Reference list must be deleted, as it has no relevance to the research topic. 

While changing the style, please ensure referencing guidelines are closely following

Thanks

Author Response

Thank you very much for your comments. We have revised the references accordingly.